# Patterns of Chromosomal Instability and Clonal Heterogeneity in Luminal B Breast Cancer: A Pilot Study

**DOI:** 10.3390/ijms25084478

**Published:** 2024-04-19

**Authors:** Valentina Camargo-Herrera, Giovanny Castellanos, Nelson Rangel, Guillermo Antonio Jiménez-Tobón, María Martínez-Agüero, Milena Rondón-Lagos

**Affiliations:** 1School of Biological Sciences, Universidad Pedagógica y Tecnológica de Colombia, Tunja 150003, Colombia; laura.camargo07@uptc.edu.co (V.C.-H.).; giovanny.castellanos@uptc.edu.co (G.C.); 2Departamento de Nutrición y Bioquímica, Facultad de Ciencias, Pontificia Universidad Javeriana, Bogotá 110231, Colombia; rangeljne@javeriana.edu.co; 3Laboratorio de Patología, Hospital Universitario Mayor-Méderi, Bogotá 110311, Colombia; guillermoa.jimenez@urosario.edu.co; 4Grupo BIOmedUR, Escuela de Medicina y Ciencias de la Salud, Universidad del Rosario, Bogotá 110231, Colombia; 5Centro de Investigaciones en Microbiología y Biotecnología-UR (CIMBIUR), Facultad de Ciencias Naturales, Universidad del Rosario, Bogotá 110231, Colombia

**Keywords:** luminal B breast cancer, chromosomal instability, clonal heterogeneity, clinical outcomes, risk stratification

## Abstract

Chromosomal instability (CIN), defined by variations in the number or structure of chromosomes from cell to cell, is recognized as a distinctive characteristic of cancer associated with the ability of tumors to adapt to challenging environments. CIN has been recognized as a source of genetic variation that leads to clonal heterogeneity (CH). Recent findings suggest a potential association between CIN and CH with the prognosis of BC patients, particularly in tumors expressing the epidermal growth factor receptor 2 (HER2+). In fact, information on the role of CIN in other BC subtypes, including luminal B BC, is limited. Additionally, it remains unknown whether CIN in luminal B BC tumors, above a specific threshold, could have a detrimental effect on the growth of human tumors or whether low or intermediate CIN levels could be linked to a more favorable BC patient prognosis when contrasted with elevated levels. Clarifying these relationships could have a substantial impact on risk stratification and the development of future therapeutic strategies aimed at targeting CIN in BC. This study aimed to assess CIN and CH in tumor tissue samples from ten patients with luminal B BC and compare them with established clinicopathological parameters. The results of this study reveal that luminal B BC patients exhibit intermediate CIN and stable aneuploidy, both of which correlate with lymphovascular invasion. Our results also provide valuable preliminary data that could contribute to the understanding of the implications of CIN and CH in risk stratification and the development of future therapeutic strategies in BC.

## 1. Introduction

BC is a prevalent illness, standing as one of the foremost health challenges globally. It is estimated that this neoplasia constitutes approximately 22.9% of cancers in women, emerging as one of the leading causes of cancer-related death in this population group [1]. Moreover, BC stands out as the most prevalent type of cancer, with the highest mortality rates, reaching approximately 2.3 million cases in 2020 [2]. The molecular categorization of BC is based on the presence of four clinically standardized biomarkers: estrogen (ER) and progesterone (PR) hormone receptors, epidermal growth factor receptor 2 (HER2), and the cell proliferation marker Ki67. These biomarkers allow the classification of this neoplasm into at least four subgroups based on their presence or absence [3,4], including luminal A and luminal B, HER2 enriched and basal-like. Each subgroup carries a different prognosis [5]. 

Luminal B tumors account for 15–20% of BC cases. They display a more aggressive phenotype, elevated histological grade, higher proliferative index, and an unfavorable prognosis [6,7]. This subtype exhibits an increased rate of recurrence and reduced survival rates following relapse in comparison to the luminal A subtype [8]. According to the 2013 St. Gallen Consensus [9], the luminal B BC subtype can be further classified into luminal B-like HER2 negative (HER2−) and luminal B-like HER2 positive (HER2+). Luminal B-like HER2− is characterized by its positivity for ER, negativity for HER2, and the presence of at least one of the following features: high Ki67 expression and negativity or low expression of PR. Conversely, the luminal B-like HER2+ is characterized by its positivity for ER, overexpression or amplification of HER2, with any level of Ki67, and any PR expression. Therefore, understanding the pattern of recurrence and clinical outcomes of the luminal B BC subtype is crucial. Although technological advances have led to a greater understanding of luminal B BC as a heterogeneous disease, present immunohistochemical, clinicopathological, and molecular markers continue to leave a significant portion of patients with the possibility of excessive or insufficient treatment. Thus, the identification and standardization of new, easily accessible prognostic and predictive markers is imperative. These markers could provide additional information to improve the stratification of cancer risk, predict clinical outcomes, and guide future therapeutic strategies for luminal B BC patients. A promising prognostic and predictive marker is CIN, a prevalent characteristic in solid tumors. Given that BC is characterized by unstable karyotypes and the fact that disease outcome and therapy response may depend on such instability, defining the level of CIN and CH could carry significant implications for the stratification of risk and the development of future therapeutic strategies in BC. 

CIN is recognized as a distinctive characteristic of cancer [10], where the loss or gain of complete chromosomes (aneuploidy) or chromosomal fragments is characteristic [11]. Aneuploidy can be stable or unstable. Stable aneuploidy is characterized by the presence of the same type of numerical alterations in the majority of cells. Conversely, unstable aneuploidy is characterized by presenting cell-to-cell variability in the number of chromosomes and is, therefore, a source of karyotypic heterogeneity [12]. In fact, unstable aneuploidy promotes the simultaneous growth of diverse tumor subpopulations, promoting both inter- and intratumoral genomic heterogeneity [11,13]. CIN promotes intratumoral heterogeneity, allowing cancer cells to adapt to environmental stress. At the same time, it promotes the development of more aggressive cancer cells, thus contributing to treatment resistance [14,15]. The significance of CIN and CH in therapy response lies in their capacity to trigger gene regulatory interactions and alter protein concentrations. Both these factors have the potential to influence cell responses to drug treatments [16]. In this context, it has been proposed that chromosomal alterations in individual cancer cells may give rise to diverse drug sensitivities, thereby fostering the survival of a subset of the tumor cell population [17]. Another fact that remains unknown is whether CIN in luminal B BC tumors, above a specific threshold, could have a detrimental effect on the growth of human tumors or whether low or intermediate CIN levels could be associated with a more favorable prognosis for luminal B BC patients compared with elevated levels. Considering the above, the objective of this study was to assess CIN and CH in tumor tissue samples from ten patients with luminal B BC and compare them with established clinicopathological parameters. Evaluating CIN and CH in luminal B BC could improve cancer risk stratification, the prediction of clinical outcomes, and future therapeutic approaches.

## 2. Results

### 2.1. Patients and Clinicopathological Data

The clinical and pathological characteristics of the patients are described in Table 1. All cases were free of metastasis at the time of diagnosis. Four cases (40%) were ER+/PR+/HER2−, three (30%) were ER+/PR−/HER2−, and three (30%) were ER+/PR+/HER2+.

### 2.2. CIN and CH Levels

Based on the CIN level (% CIN), each patient was categorized as having low CIN (CIN = 0–25%), intermediate CIN (CIN = 26–50%), high CIN (CIN = 51–70%), or extreme CIN (CIN > 70%) [18]. The findings of this study reveal that most patients (90%) presented intermediate CIN, with values ranging between 33% and 49%, while only 1 (10%) patient (ER+/PR+/HER2+) presented high CIN (52%) (Figure 1 and Figure 2 and Appendix A).

Once the possible existence of variations in the level of CIN according to PR and HER2 status was determined, no important variations were observed. However, the highest CIN ranges occurred in ER+/PR+/HER2− patients (Figure 3A and Appendix A). Notably, lymphovascular invasion appeaed to be significant in patients with intermediate CIN. 

All patients (100%) exhibited elevated CH, with values ranging between 2.5 and 3.4. There were no variations in the level of CH according to PR and HER2 status (Figure 3B and Appendix A). 

The CIN level for each chromosome was also determined. The results of this study highlight that the most stable chromosomes were chromosomes 11 (CEP11) and 15 (CEP15), while the least stable chromosomes were chromosomes 8 (CEP8) (CIN = 50%), 3 (CEP3) (CIN = 49%), and 17 (CEP17) (CIN = 49%) (Figure 4).

### 2.3. CEP Copy Number Variations (CNVs)

CNVs (gains and losses) for CEP2, CEP3, CEP8, CEP11, CEP15, and CEP17 were evaluated in all cases. Conforming to the criteria previously established in defining chromosomal gains and losses, a chromosome was deemed to exhibit gains when the mean count of centromeric signals was equal to or greater than 3 (CEP ≥ 3) and losses when the mean count of centromeric signals was less than 1.6 (CEP < 1.6) [19].

A copy number gain was observed only for CEP17 in 1 (10%) case (ER+/PR+/HER2−) (Appendix A). While copy number losses for CEP2, CEP3, CEP8, CEP11, CEP15, and CEP17 were observed in 7 (70%), 6 (60%), 9 (90%), 10 (100%), 9 (90%), and 7 (70%) cases, respectively (Figure 2). These results show that losses were more frequent than gains.

### 2.4. Determination of Stable and Unstable Aneuploidy

Stable or unstable aneuploidy was determined for each chromosome and for each BC patient according to PR and HER2 status. The chromosomes with the most stable aneuploidy were chromosomes 11 (CEP11) and 15 (CEP15), while the chromosomes with the least stable aneuploidy were chromosomes 2 (CEP2), 3 (CEP3), 8 (CEP8), and 17 (CEP17) (Figure 5).

Regarding the assessment of aneuploidy stability in luminal B BC patients, it was observed that all patients, irrespective of PR and HER2 status (ER+/PR+/HER2−, ER+/PR−/HER2−, and ER+/PR+/HER2+), exhibited stable aneuploidy. This stability was characterized by stability in the number of copies of chromosomes 11 and 15. Among the chromosomes exhibiting less stable aneuploidy were chromosome 2 for ER+/PR+/HER2− patients and chromosome 3 for ER+/PR−/HER2− and ER+/PR+/HER2+ patients (Figure 6).

### 2.5. Association between CEP Copy Number Variations and CH with Clinicopathological Parameters

To assess the associations between CNVs per chromosome and clinicopathological characteristics, a multivariate analysis with Pearson correlation coefficient was performed. A positive correlation was found between CNVs of chromosome 2 with T stage (0.60), between CNVs of chromosome 3 with N stage (0.89), and between CNVs of chromosome 8 with lymphovascular invasion (0.56) (Figure 7). No correlations were found between CNVs of chromosomes 2, 3, 8, 11, 15, and 17, with PR, HER2, and Ki67, nor among CH with any of the clinicopathological characteristics studied (Ht, T, N, LI, PR, HER2, and Ki67) (Appendix A).

## 3. Discussion

The clinical outcome and therapeutic decision regarding luminal B BC patients are mainly based on clinicopathological parameters. Specifically, tumor size, histologic grade, histological type, immunohistochemical results of prognostic factors, and lymph node status play an important role in cancer risk stratification. However, despite the success of this approach, there is still a risk of over- or under-treating certain patients. The above underscores the need to conduct cancer risk stratification by integrating clinicopathological parameters with enhanced insights into molecular-level alterations. This integration could improve the prediction of clinical outcomes and inform future therapeutic strategies for luminal B BC patients. Given that CIN is a defining characteristic of BC and that risk stratification and prognosis definition may depend on this instability, defining the level of CIN and CH could have significant implications for risk stratification and the development of future therapeutic strategies in BC. The above, in turn, could facilitate the optimization of BC diagnosis and prognosis. 

The results of this study reveal that luminal B BC patients exhibit intermediate CIN and stable aneuploidy, with chromosomal losses being more frequent than gains. CIN, CH, and aneuploidy represent distinct and crucial characteristics of tumor cell populations, contributing to tumor evolution with significant implications for prognosis and therapy response. In fact, statistically significant correlations were observed between moderate CIN and lymphovascular invasion. 

The findings of this study are in line with previous research, which suggests that ER+ tumors exhibit the lowest average CIN70 score [18]. CIN70 is a method used to determine the level of CIN from the expression of 70 specific genes consistently associated with aneuploidy in populations of tumor samples [20]. However, this method evaluates the level of CIN from a pool of cells rather than assessing CIN cell by cell, as was done in this study using FISH with six centromeric probes. 

Aneuploidy, potentially induced by CIN [21], is among the most prevalent abnormalities in cancer. Therefore, identifying chromosomes associated with CIN may potentially improve risk stratification and optimize outcome prediction. In this regard, we found that in luminal B BC patients, chromosomes 11 and 15 exhibit the most stable aneuploidy, while chromosomes 2, 3, and 8 were the least stable. These results are significant, as they suggest that determining the state of aneuploidy (stable or unstable) could enable the prediction of clinical outcomes in BC patients. Indeed, it has been indicated that individuals with genetically stable tumors exhibit a more favorable disease outcome compared to those with unstable tumors [21]. In this sense, our results suggest that intermediate CIN could be associated with clinical characteristics of intermediate prognosis in luminal B BC patients. This is supported by the statistically significant correlation observed between stable aneuploidy (losses) of chromosomes 2, 3, and 8 with the T stage (T1 and T2), N stage (N1 and N2), and lymphovascular invasion, respectively. 

CNVs were identified in all evaluated chromosomes, with losses being more frequent than gains. For instance, the gain of chromosome 17 (CEP17) was observed in only one patient (ER+/PR+/HER2−). This finding aligns with prior research suggesting that, among the associated copy number alterations, the gain of chromosome 17 is a characteristic feature of luminal B BC [22]. Even though copy number gains in CEP17 have been linked to adverse clinical outcomes [19,23,24] and the response to chemotherapy with anthracyclines in BC patients [25,26], the relevance of a copy number gain in CEP17 remains unclear. The losses observed across all analyzed chromosomes are noteworthy, particularly due to the presence of tumor suppressor genes (TSGs) known for their pivotal roles in monitoring and repairing double-strand breaks (DSBs) and maintaining genome stability (Table 2). In fact, CNVs in these genes have been associated with an increased risk of genomic instability and cancer development [27,28]. 

Notably, losses of chromosome 8 were observed in 90% of luminal B BC patients, which showed a statistically significant correlation with lymphovascular invasion. These results suggest that CEP8 copy number loss (indicative of CIN) could be a predictor for poor prognosis in luminal B BC patients. Indeed, some studies have shown that 8p loss is closely linked to a subset of breast cancers characterized by high aggressiveness and genetic instability, attributing to this chromosome a possible role in the regulation of genomic stability [27]. The observed statistically significant correlation between losses in chromosome 8 and unfavorable tumor phenotype characteristics suggests that this chromosome harbors one or more genes that are crucial to maintaining genome stability. Consequently, alterations in chromosome 8 could potentially contribute to the development of more aggressive cancer phenotypes and alterations in the response to therapy. Indeed, a recent study suggested that sub-clonal CNAs favor the positive selection of oncogenes and the negative selection of tumor suppressor genes (TSGs) [28]. In fact, several candidate TSGs in BC located on chromosome 8p have been identified (Table 2). 

The identification of CIN and losses on chromosome 8 (losses associated with lymphovascular invasion) could serve as a pivotal pathway for the development of novel therapeutic strategies. Validation of these findings in a larger cohort of samples could provide crucial insights into targeted therapeutic approaches for improved BC management. Further, understanding the specific genes residing in chromosome 8 and their roles in maintaining genome stability can unravel potential targets for precision therapies. 

A limitation of this study involved restricted access to samples. However, these ten patients represent a subset of the tumoral subtype within the BC population, providing valuable insights into the role of CIN and CH in luminal B BC. Furthermore, the results of this pilot study serve as a basis for future research with larger cohorts, enabling the validation of the findings and the development of future therapeutic strategies in BC. Other factors contributing to the limited sample size include, firstly, negative responses from some patients who refused to participate in the study. Secondly, in Colombia, where the study was undertaken, gaining access to tumor samples is challenging, as laws allowing such access for research purposes are still in the early stages of consideration. 

## 4. Materials and Methods

### 4.1. Patients and Clinicopathological Data

Archival formalin-fixed paraffin-embedded (FFPE) tumor tissue was obtained from ten (10) luminal B BC patients diagnosed at Hospital Universitario Mayor—Méderi, Bogotá, Colombia, between 2000 and 2018. Sections of these primary tumor tissues were selected from representative tumor areas by an expert pathologist. Clinicopathological data were obtained from medical records and sections stained with hematoxylin and eosin. Data on the primary tumor included age, the breast affected by cancer (right or left), histologic subtype (Ht), T stage, N stage, lymphovascular invasion, HER2 status, progesterone, and estrogen receptor status, Ki67 index, and luminal B types. The main inclusion criteria encompassed individuals diagnosed with luminal B BC between 2000 and 2018, treated at the Hospital Universitario Mayor—Méderi for minimum of five (5) years, undergoing hormonal therapy and/or chemotherapy, with and without relapse. Exclusion criteria comprised individuals who have received neo-adjuvant or adjuvant therapy (chemotherapy, radiotherapy, hormonal therapy) prior to sample collection and those with a current diagnosis of other types of cancer. The research received approval from the Research Ethics Committee at Universidad del Rosario in Bogotá, Colombia. Informed consent was obtained from all individual participants included in the study.

### 4.2. Dual-Color Fluorescence In Situ Hybridization (FISH) Assays

Dual-color FISH assays were carried out on FFPE tissue sections of 4 μm by using centromeric probes (CEP) for chromosomes 2, 3, 8, 11, 15, and 17 (Cytocell. Cytocell Ltd., Cambridge, UK). CEP for chromosomes 2, 3, and 8 (CEP2, CEP3, and CEP8) were labeled with spectrum red, while CEP for chromosomes 11, 15, and 17 (CEP 11, CEP15, and CEP17) were labeled with spectrum green. Three different dual-color FISH assays were performed using the following combination of probes: CEP2 and CEP11, CEP3 and CEP15, and CEP8 and CEP17. Ten randomly selected areas of the FFPE tissues slides were selected and captured using an Olympus microscope with Cytovision System 7.4 cytogenetic software (Leica Biosystems Richmond, Inc., Richmond, IL, USA). Chromosomes 2 and 15 were chosen because alterations in their copy numbers are infrequent in BC [40]. While chromosomes 3, 8, 11, and 17 were selected because they commonly undergo alterations in BC [41]. 

### 4.3. Evaluation of CIN and CH

One hundred non-overlapping nuclei were examined for each chromosome and for each patient. CIN was established in accordance with the method specified by Lengauer et al. (1997) [42]. Briefly, the CIN level for each patient was determined as follows: first, for each individual chromosome, the percentage of nuclei with a number of centromeric signals (CEP) different from the most frequent number of signals in the tumor cell population (modal number) was calculated. After establishing the CIN level (% CIN) for each individual chromosome, the mean CIN level was computed for the six analyzed chromosomes [42,43]. According to the CIN level (% CIN), each patient was classified as having low CIN (CIN = 0–25%), intermediate CIN (CIN = 26–50%), high CIN (CIN = 51–70%), or extreme CIN (CIN > 70%) [18]. 

CH was determined by calculating the true diversity index (TD) using order 1 (q = 1), which integrates the Shannon index and is represented as follows: NEE = e^H, where NEE corresponds to the effective number of species, e is equal to Euler or Napier’s constant, and H is the Shannon index. Shannon index combines both the quantity and prevalence of cell clones within each patient, adhering to established methods described in previous publications [40,44]. According to the CH level, each patient was classified as having low CH (CH < 1.5), intermediate CH (CH > 1.5 and <2), or high CH (CH > 2).

### 4.4. Evaluation of CEP Copy Number Variations (CNVs)

For each of the six analyzed chromosomes (2, 3, 8, 11, 15, and 17), the centromeric signal (CEP probe) count was determined in 100 non-overlapping tumor nuclei. Subsequently, the mean CEP signal was calculated for each of the six chromosomes individually. Based on the mean of CEP signals, each chromosome was classified as having gains if the mean CEP signal count was ≥3 [19] or losses if the mean CEP signal count was <1.6 [45].

### 4.5. Determination of Stable and Unstable Aneuploidy

Given that CIN can classify stable aneuploid tumor cells (populations of tumor cells in which almost all cells have the same chromosome number) along with unstable aneuploid tumor cells (populations of tumor cells in which most cells have varying numbers of chromosomes) [40], we decided to ascertain whether the type of aneuploidy present in each chromosome was stable or unstable. Stable or unstable aneuploidy was also determined for each BC patient according to PR and HER2 status (ER+/PR+/HER2−, ER+/PR−/HER2−, and ER+/PR+/HER2+). Stable or unstable aneuploidy was calculated by assessing the proportion of cells with identical probe signal patterns. A chromosome and a BC patient were considered to exhibit stable aneuploidy if more than 20% (≥20%) of the cells showed identical probe signal patterns, while a BC patient or chromosome with less than 20% (<20%) of the cells showing identical probe signal patterns was classified as having unstable aneuploidy [46].

### 4.6. Data Analysis

Descriptive statistics were used to assess the distribution and frequency of clinicopathological variables, CIN and CH, in patients with luminal B BC. The normality of the data was evaluated using the Shapiro–Wilk test. Correlations among CIN and CH levels with clinicopathological variables were established using the Pearson test for normally distributed data and Spearman and Kruskal–Wallis tests for non-parametric data. To establish associations among CIN and CH with categorical clinicopathological variables, contingency tables and measures of association, such as Cramer’s contingency coefficient, were performed. Subsequently, to evaluate the existence of statistically significant differences between luminal B BC patients according to PR and HER2 status (ER+/PR+/HER2−, ER+/PR+/HER2+, ER+/PR−/HER2−), a test of difference between parametric and non-parametric means was performed using analysis of variance (ANOVA). Finally, in order to evaluate the association between CNVs per chromosome with clinicopathological variables, a multivariate analysis was performed using the Pearson correlation coefficient. All statistical analyses were performed using R Studio version 4.0.2. *p*-values of less than 0.05 (*p* < 0.05) were considered statistically significant.

## 5. Conclusions

The results of this pilot study reveal that luminal B BC patients exhibit intermediate CIN and stable aneuploidy, both of which correlate with lymphovascular invasion. The evaluation of CIN and CH levels holds potential value for validation with a larger cohort of patients and for their potential inclusion as prognostic or predictive biomarkers in luminal B BC. Despite the limited number of samples used, this study provides valuable contributions to the field of breast cancer research, as well as valuable preliminary data that lays the foundation for future research in this area.

## Figures and Tables

**Figure 1 ijms-25-04478-f001:**
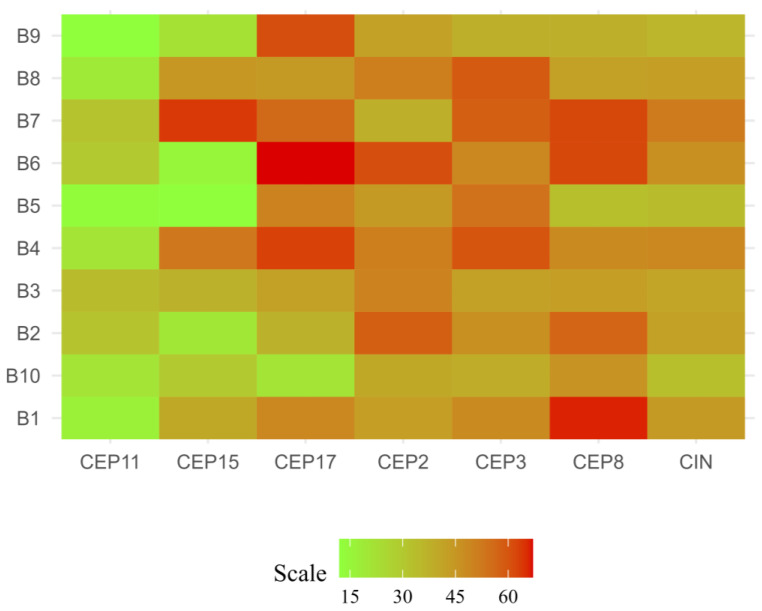
Chromosomal instability (CIN) observed in luminal B BC patients. Each row represents a patient, while each column represents a chromosome. The letter B and its respective number, correspond to the codes assigned to each patient. The level of CIN is color-coded according to the legend at the bottom.

**Figure 2 ijms-25-04478-f002:**
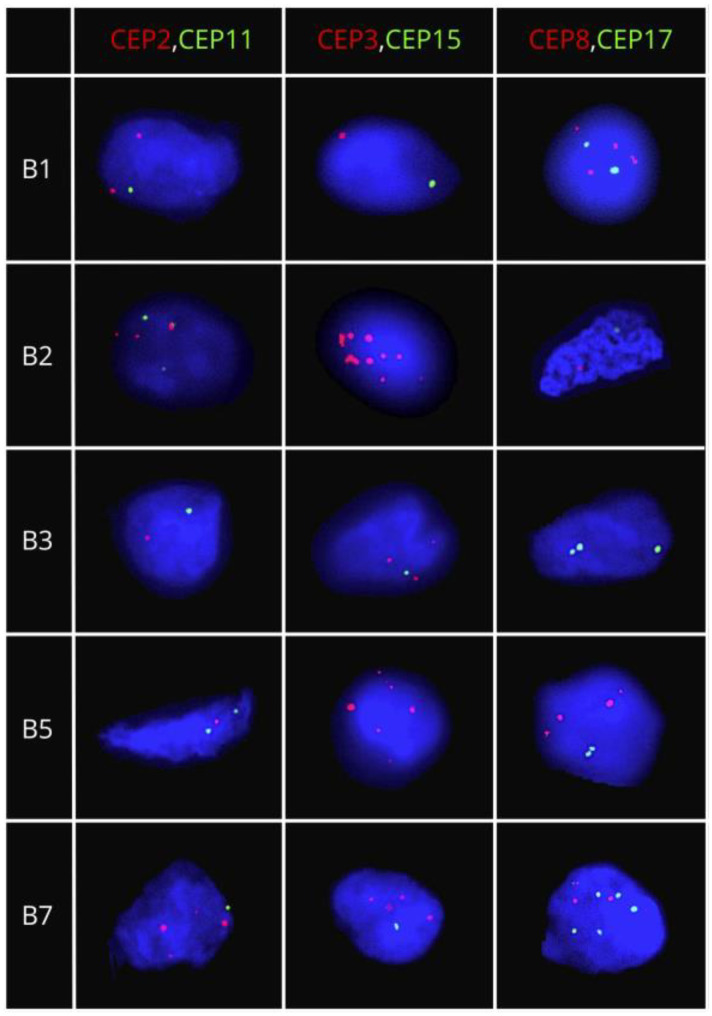
Representative FISH images for luminal B BC patients. Dual-color FISH was performed on nuclei spreads for chromosomes 2 and 11, chromosomes 3 and 15, and chromosomes 8 and 17, using centromeric probes (CEP) labeled with different spectrum colors: spectrum red for CEP2, CEP3, and CEP8, and spectrum green for CEP 11, CEP15, and CEP17. The letter B and its respective number, correspond to the codes assigned to each patient.

**Figure 3 ijms-25-04478-f003:**
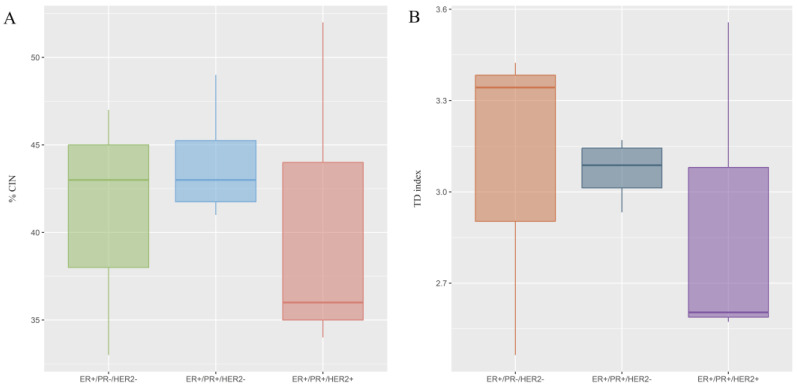
Chromosomal instability (CIN) and clonal heterogeneity (CH) observed in luminal B BC patients according to PR and HER2 status. (**A**) The level of CIN was classified as having low CIN (CIN = 0–25%), intermediate CIN (CIN = 26–50%), high CIN (CIN = 51–70%), or extreme CIN (CIN > 70%). (**B**) CH was determined by true diversity (TD). Values below 1.5 were considered indicative of low CH, values between 1.6 and 2 were considered indicative of intermediate CH, and values higher than 2 were considered indicative of high CH.

**Figure 4 ijms-25-04478-f004:**
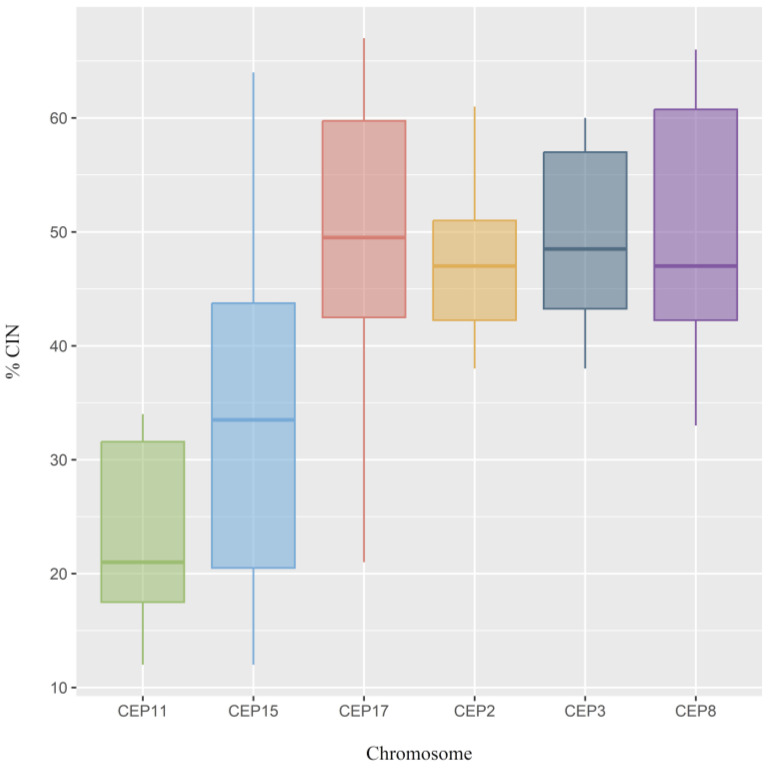
Chromosomal instability (CIN) observed in all chromosomes analyzed in luminal B BC patients. Each chromosome was classified as having low CIN (CIN (CIN = 0–25%), intermediate CIN (CIN = 26–50%), high CIN (CIN = 51–70%), or extreme CIN (CIN > 70%).

**Figure 5 ijms-25-04478-f005:**
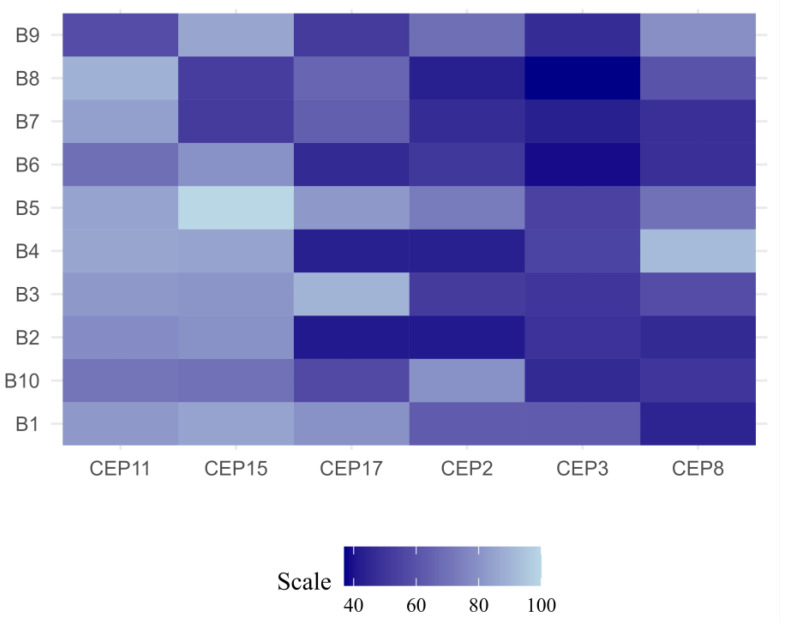
Stage of the aneuploidy observed in luminal B BC patients. Each row represents a patient, while each column represents a chromosome. The stability of the aneuploidy is color-coded according to the legend at the bottom. The letter B corresponds to the code given to each patient. Values ≥20% were considered as stable aneuploidy, while values <20 were considered as unstable aneuploidy.

**Figure 6 ijms-25-04478-f006:**
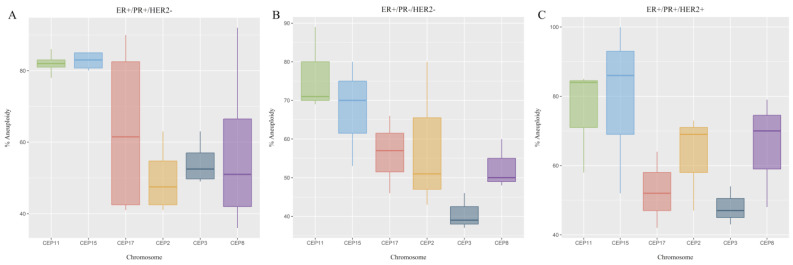
Stability of the aneuploidy in luminal B BC patients according to PR and HER2 status. (**A**) ER+/PR+/HER2−, (**B**) ER+/PR−/HER2−, and (**C**) ER+/PR+/HER2+. Values ≤ 20 were considered unstable, while values >20 were considered stable.

**Figure 7 ijms-25-04478-f007:**
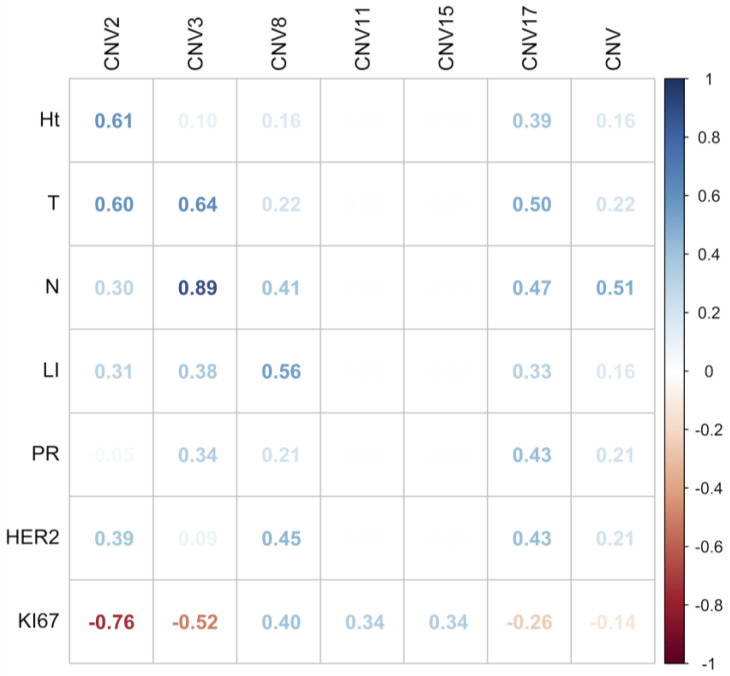
Multivariate analysis with Pearson correlation coefficient for copy number variations of chromosomes 2, 3, 8, 11, 15, and 17 and clinicopathologic characteristics. Values greater than 0.5 are indicative of a statistically significant correlation. Abbreviations: Ht, Histotype; T, tumor size; N, lymph nodes, LI; lymphovascular invasion; PR, progesterone receptor.

**Table 1 ijms-25-04478-t001:** Clinicopathological characteristics of luminal B BC patients.

Clinicopathological Characteristics		CIN		*p*-Value	CH	*p*-Value
		Intermediate	High		High	
Total Number of patients		9 (0.9)	1 (0.1)	-	10 (1)	-
Age	≥50 years	9 (0.9)	1 (0.1)	0.187	10 (1)	0.467
Breast	Right	6 (0.6)	1 (0.1)	0.788	7 (0.7)	1
	Left	3 (0.3)	0		3 (0.3)	
Histological type	Invasive ductal carcinoma	7 (0.7)	1 (0.1)	0.598	8 (0.8)	1
	Mixed carcinoma	2 (0.2)	0		2 (0.2)	
Histologic grade	ll	7 (0.7)	1 (0.1)	0.598	8 (0.8)	1
	lll	2 (0.2)	0		2 (0.2)	
T stage	T1	2 (0.2)	0	0.788	2 (0.2)	1
	T2	6 (0.6)	1 (0.1)		7 (0.7)	
	T3	1 (0.1)	0		1 (0.1)	
N stage	N0	0	1 (0.1)	0.644	1 (0.1)	1
	N1	3 (0.3)	1 (0.1)		4 (0.4)	
	N2	3 (0.3)	0		3 (0.3)	
	N3	2 (0.2)	0		2 (0.2)	
Lymphovascular invasion	Absent	1 (0.1)	1 (0.1)	0.035 *	2 (0.2)	1
	Present	8 (0.8)	0		8 (0.8)	
Progesterone receptor	Positive	6 (0.6)	1 (0.1)	0.49	7 (0.7)	1
	Negative	3 (0.3)	0		3 (0.3)	
HER2 status	Positive	2 (0.2)	1 (0.1)	0.107	3 (0.3)	1
	Negative	7 (0.7)	0		7 (0.7)	
Ki67 index	≥20%	9 (0.9)	1 (0.1)	0.661	10 (1)	0.467
Receptor status	ER+/PR+/HER2−	4 (0.4)	0	0.791	4 (0.4)	0.92
	ER+/PR+/HER2+	2 (0.2)	1 (0.1)		3 (0.3)	
	ER+/PR−/HER2−	3 (0.3)			3 (0.3)	

* Statistically significant difference relative to clinicopathological characteristics at *p* ≤ 0.05 (Pearson test).

**Table 2 ijms-25-04478-t002:** Tumor suppressor genes located on the chromosomes analyzed in this study.

Gene	Location	Function	References
*MSH2*	2p22	Mismatch repair	[29]
*RARβ2*	3p24	Retinoic acid receptor	[29]
*PRKCD*	3p21.1	Involved in DNA damage response	[30]
*BAP1*	3p21.1	Regulate ubiquitination during the DNA damage response and the cell cycle	[31]
*MLH1*	3p21	Mismatch repair	[29]
*FHIT*	3p14.2	Plays an important role in the regulation of apoptosis	[31]
*PMC1*	8p22	Encoded for a protein essential for anchoring microtubules to the centrosome	[32]
*DLC1*	8p22	Promoter of apoptosis	[33,34]
*MTUS1*	8p22	Slows down mitotic progression by prolonging metaphase	[35]
*LZTS1*	8p21	Inhibits the Cdk1/cyclin B1 complex	[36]
*WRN*	8p12	Critical controller of fragile site stability, essential for preserving genome stability	[37]
*ATM*	11q22.3	DNA repair	[31]
*MIR34B*	11q23.1	Involved in DNA damage response	[38]
*TP53*	17p13.1	DNA repair, cell cycle, apoptosis, and angiogenesis regulator	[31]
*BRCA1*	17q21.31	DNA repair; maintains genomic stability	[39]
*BRIP1*	17q23.2	Important in the normal double-strand break repair function of breast cancer	[31]

## Data Availability

The original contributions presented in the study are included in the article/Appendix A; further inquiries can be directed to the corresponding authors.

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
