# Peer review of "Patterns of Chromosomal Instability and Clonal Heterogeneity in Luminal B Breast Cancer: A Pilot Study"

_ijms, 2024, doi:10.3390/ijms25084478_

Round 1

Reviewer 1 Report

Comments and Suggestions for Authors

It's a well-written paper that shows significant outcomes for the stratification of BC and its treatment.

1. In the methods section the authors can explain the process of CIN and CH and evaluation of CEP copy number. (4.3, 4.4)

2. The authors should mention what is the significance of checking chromosome 2, 3, 8, 11, 15, and 17.

3. The authors can provide the details of possible genes related to breast cancer present on these chromosomes, which can impact disease progression and treatment outcomes.

4. Sample size is very limited. Based on this sample size, it is difficult to imply that these methods can be used for disease stratification and prediction of treatment.  

Comments on the Quality of English Language

English quality is good. In a few places the sentence structure can be improved to avoid repetition of words for example page 2, last paragraph lines 1,2.

Author Response

Tunja, February 9 - 2024

Dear
Dr. Elena Levantini
Special Issue Editor
International Journal of Molecular Science 

Please find enclosed the revised version of the research article entitled “Patterns of chromosomal instability and clonal heterogeneity in Luminal B breast cancer: a pilot study”, by Camargo and co-workers.

We have accepted all suggestions, as specified by a point-by-point response, as requested. All amendments have been highlighted in blue color in the manuscript file. We acknowledge that the revision following the raised criticisms highly improved the quality of the manuscript.

We thank you for your attention and hope that this work may now fulfil the scientific standards of International Journal of Molecular Science.

Sincerely yours,

Milena Rondón-Lagos
Universidad Pedagógica y Tecnológica de Colombia. UPTC
Tunja, Colombia
sandra.rondon01@uptc.edu.co
milerondon@yahoo.com

RESPONSES TO REVIEWER 1

It's a well-written paper that shows significant outcomes for the stratification of BC and its treatment. 
AUTHORS: We thank the reviewer for finding our article well written and with significant results for the stratification of BC and its treatment.

1. In the methods section the authors can explain the process of CIN and CH and evaluation of CEP copy number (4.3, 4.4).
AUTHORS: We appreciate the reviewer's suggestion. Indeed, the methodology for assessing chromosomal instability (CIN) and chromosomal heterogeneity (CH) is detailed in section 4.3 of the manuscript. Similarly, the methodology for evaluating the CEP copy number is outlined in section 4.4. We trust that we have addressed the reviewer's request by incorporating the requested information as indicated. 

2. The authors should mention what is the significance of checking chromosome 2, 3, 8, 11, 15, and 17. 
AUTHORS: We thank the reviewer for this pertinent suggestion. The importance of evaluating the copy number of chromosomes 2, 3, 8, 11, 15 and 17 is included in section 4.2. [Dual color Fluorescence in Situ Hybridization (FISH) assays].

3. The authors can provide the details of possible genes related to breast cancer present on these chromosomes, which can impact disease progression and treatment outcomes. 
AUTHORS: Following the reviewer's recommendation, we have incorporated details of potential genes associated with breast cancer present on the analyzed chromosomes, into Table 2.

4. Sample size is very limited. Based on this sample size, it is difficult to imply that these methods can be used for disease stratification and prediction of treatment. 
AUTHORS: We appreciate the reviewer's suggestion. Recognizing that our results were derived from a sample size of ten patients, we have rewritten the last  paragraph of the discussion highlighting that our study is a pilot study that provides valuable preliminary data that could contribute to the understanding of the implications of CIN and CH in risk stratification and development of future therapeutic strategies in BC. Likewise, we have consistently used cautious language throughout the paper, such as "our results suggest that" or "our results may indicate," among others. In order to highlight the potential use of our results and the importance of validating them in future studies with a larger number of patients, we have made adjustments to the title, discussion and conclusions. 

In future investigations, we aspire to expand our study cohort, encompassing not only more patients with luminal B breast cancer but also with those representing other tumor subtypes.

English quality is good. In a few places the sentence structure can be improved to avoid repetition of words for example page 2, last paragraph lines 1,2. 
AUTHORS: We thank the reviewer for this pertinent suggestion. We have revised sentence structure throughout the manuscript to minimize word repetition. In addition,  
our manuscript has undergone English review and editing to ensure clarity, coherence, and grammatical accuracy, as can be verified in the attached certificate.

Reviewer 2 Report

Comments and Suggestions for Authors

In their manuscript, Camargo-Herrera et al present the patterns of chromosomal instability and clonal heterogeneity in luminal B breast cancer. The subject that the authors chosed is of actuality, because, as it is already known, BC is still one of the most prevalent cancer worldwide. The research team tried to use the assessment of CIN as a tool for risk stratification in luminal B BC patients. Even though this idea might be of interest for the current research in this specific domain, in my opinion, this study MUST be improoved before being taken into consideration for publication. Firstly, the authors included only 10 patients, a number that is extremely small, reason why the statistical significance of the results is doubtful. Secondly, the study design is flawed because is lacks a control group, or a group of patients with other than luminal B BC as comparison. In lack of those, the authors cannot conclude that that type of genetic modifications are solely specific for luminal B type.

To conclude, my opinion is that this study, even though it has potential, needs to be improoved in study design and regarding the number of patients included, before being takne into consideration for publication.

Comments on the Quality of English Language

The English also needs minor revisions. The authors should, after impooving the study design and send it again for peer reviewing, have an english proofreading certificate.

Author Response

Tunja, February 9 - 2024

Dear
Dr. Elena Levantini
Special Issue Editor
International Journal of Molecular Science 

Please find enclosed the revised version of the research article entitled “Patterns of chromosomal instability and clonal heterogeneity in Luminal B breast cancer: a pilot study”, by Camargo and co-workers.

We have accepted all suggestions, as specified by a point-by-point response, as requested. All amendments have been highlighted in blue color in the manuscript file. We acknowledge that the revision following the raised criticisms highly improved the quality of the manuscript.

We thank you for your attention and hope that this work may now fulfil the scientific standards of International Journal of Molecular Sciences.

Sincerely yours,

Milena Rondón-Lagos
Universidad Pedagógica y Tecnológica de Colombia. UPTC
Tunja, Colombia
sandra.rondon01@uptc.edu.co
milerondon@yahoo.com

RESPONSES TO REVIEWER 2

In their manuscript, Camargo-Herrera et al present the patterns of chromosomal instability and clonal heterogeneity in luminal B breast cancer. The subject that the authors chosed is of actuality, because, as it is already known, BC is still one of the most prevalent cancer worldwide. The research team tried to use the assessment of CIN as a tool for risk stratification in luminal B BC patients. Even though this idea might be of interest for the current research in this specific domain, in my opinion, this study MUST be improved before being taken into consideration for publication. 

Firstly, the authors included only 10 patients, a number that is extremely small, reason why the statistical significance of the results is doubtful. 
AUTHORS: We thank the reviewer for this important observation. We acknowledge the concern regarding the limited number of patients included in our study, which consisted of ten individuals. While it's true that our sample size is small, it's important to note that the recruitment of patients with specific characteristics, such as those with specific subtypes of breast cancer, can be a challenge in our country. In our study, we focused on a subset of patients with luminal B Breast Cancer, which limited the pool of eligible participants. Despite the small sample size, our pilot study provides valuable preliminary data that contribute to the understanding of the associations between CIN and CH with prognosis and response to therapy in luminal B Breast Cancer. Additionally, we employed rigorous analyses to ensure the reliability and validity of our results within the context of the sample size. While the statistical significance of smaller sample sizes may be limited, they can still provide important insights and serve as a basis for further investigation in larger cohorts. Therefore, while we acknowledge the limitations posed by our small sample size, we believe our study adds valuable insights to the existing literature on this topic.

Secondly, the study design is flawed because is lacks a control group, or a group of patients with other than luminal B BC as comparison. In lack of those, the authors cannot conclude that that type of genetic modifications are solely specific for luminal B type. 
AUTHORS: We thank the reviewer for comments on the study design. We acknowledge your concern regarding the absence of a control group or a comparison group of patients with breast cancer subtypes other than luminal B. While we understand the importance of including such control groups for comparative analysis, our study was primarily focused on investigating chromosomal instability and heterogeneity specifically in luminal B breast cancer patients and compared them with established clinicopathological parameters. Our aim was to elucidate levels of CIN and CH for this particular subtype rather than conducting a comparative analysis across different subtypes. We recognize the value of including comparison groups in future studies to provide a more comprehensive understanding of the CIN levels across various breast cancer subtypes. We appreciate your input and will consider your suggestion for future research endeavors.

In reference to our conclusions, we sincerely apologize if we appeared to suggest that the levels of chromosomal instability and chromosomal heterogeneity identified in patients with luminal B breast cancer were solely specific to this particular tumoral subtype. Taking into account our aim, our results suggest that the luminal B breast cancer patients analyzed in this study, were characterized by presenting intermediate CIN and stable aneuploidy, which correlated with lymphovascular invasion. We regret any confusion or misinterpretation that may have arisen from our writing. To provide more clarity in this regard, we have modified the conclusions accordingly, and also, we have highlighted this point in the abstract and in the discussion.

To conclude, my opinion is that this study, even though it has potential, needs to be improved in study design and regarding the number of patients included, before being take into consideration for publication. 
AUTHORS: We thank the reviewer for the comments and for taking the time to review our study. We appreciate your insights regarding the study design and the number of patients included. We acknowledge that a larger sample size could enhance the robustness of our findings. However, it's important to note that our study was conducted with a limited sample size due to patient recruitment limitations. It's worth noting that research studies often encounter challenges in sample recruitment, and while a larger sample size would undoubtedly strengthen the study, valuable insights can still be gleaned from studies with limited sample sizes. Nonetheless, we believe that even with the limited sample size, our study provides valuable contributions to breast cancer research and valuable preliminary data that lays the foundation for future research in this area.

In order to highlight the above, we have made some adjustments to the title, discussion, and conclusions. Additionally, we emphasize in both, the discussion and conclusions sections, the importance of validating our findings in a larger patient cohort.

In future investigations, we aspire to expand our study cohort, encompassing not only more patients with luminal B breast cancer but also with those representing other tumor subtypes

The English also needs minor revisions. The authors should, after improving the study design and send it again for peer reviewing, have an English proofreading certificate. 
AUTHORS: Following the reviewer's suggestion, our manuscript has undergone English review and editing to ensure clarity, coherence, and grammatical accuracy, as can be verified in the attached certificate.

Round 2

Reviewer 1 Report

Comments and Suggestions for Authors

Revision accepted

Reviewer 2 Report

Comments and Suggestions for Authors

Even though the cohort was not enlarged, the comments the Authors added are reasonable enough, so I will suugest the publication of this manuscript.